# SymAgent: A Neural-Symbolic Self-Learning Agent Framework for Complex Reasoning over Knowledge Graphs

## Abstract

Recent advancements have highlighted that Large Language Models (LLMs) are prone to hallucinations when solving complex reasoning problems, leading to erroneous results. To tackle this issue, researchers incorporate Knowledge Graphs (KGs) to improve the reasoning ability of LLMs. However, existing methods face two limitations: 1) they typically assume that all answers to the questions are contained in KGs, neglecting the incompleteness issue of KGs, and 2) they treat the KG as a static repository and overlook the implicit logical reasoning structures inherent in KGs. In this paper, we introduce SymAgent, an innovative neural-symbolic agent framework that achieves collaborative augmentation between KGs and LLMs. We conceptualize KGs as dynamic environments and transform complex reasoning tasks into a multi-step interactive process, enabling KGs to participate deeply in the reasoning process. SymAgent consists of two modules: Agent-Planner and Agent-Executor. The Agent-Planner leverages LLM's inductive reasoning capability to extract symbolic rules from KGs, guiding efficient question decomposition. The Agent-Executor autonomously invokes predefined action tools to integrate information from KGs and external documents, addressing the issues of KG incompleteness. Furthermore, we design a self-learning framework comprising online exploration and offline iterative policy updating phases, enabling the agent to automatically synthesize reasoning trajectories and improve performance. Experimental results demonstrate that SymAgent with weak LLM backbones (i.e., 7B series) yields better or comparable performance compared to various strong baselines. Further analysis reveals that our agent can identify missing triples, facilitating automatic KG updates. The code is available at https://anonymous.4open.science/r/SymAgent/.

## CCS Concepts

• **Do Not Use This Code → Generate the Correct Terms for Your Paper**; *Generate the Correct Terms for Your Paper*.

## Keywords

Large Language Model Agent; Knowledge Graph; Self-Learning

**ACM Reference Format:**
Anonymous Author(s). 2018. SymAgent: A Neural-Symbolic Self-Learning Agent Framework for Complex Reasoning over Knowledge Graphs. In *Proceedings of Make sure to enter the correct conference title from your rights confirmation emai (Conference acronym 'XX).* ACM, New York, NY, USA, 11 pages. https://doi.org/XXXXXXX.XXXXXXX

## 1 INTRODUCTION

Knowledge Graphs (KGs) store massive factual triples in a graph-structured format, providing critical supportive information to various semantic web technologies [9, 11, 37]. Recently, Large Language Models (LLMs) have demonstrated impressive capabilities in language understanding and information integration across diverse domains [47]. However, they are limited by the lack of precise knowledge and are prone to hallucinations in their responses [40]. Given that KGs encapsulate the essence of data interconnectivity, providing explicit and explainable knowledge, integrating LLMs and KGs has garnered significant research interest. This integration facilitates a wide range of web-based applications, including search engine recommendation [18, 46], fake news detection [25], and social networks [44].

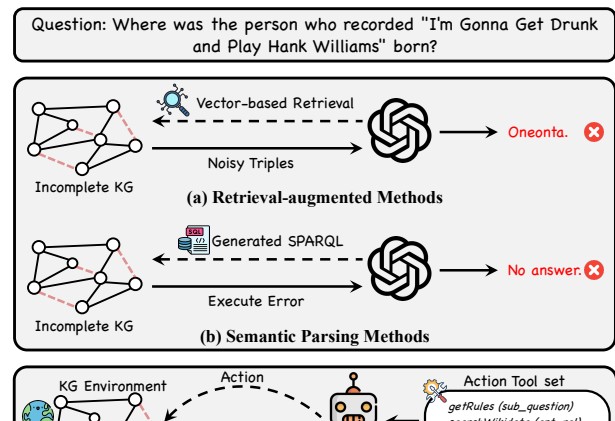

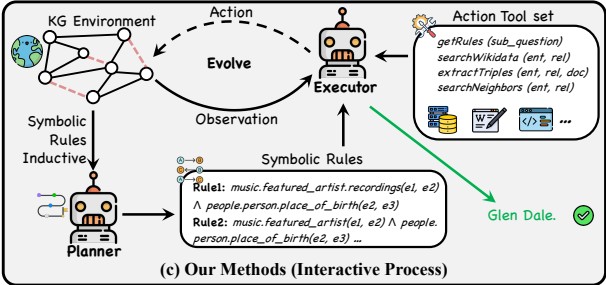

**Figure 1: Comparison between SymAgent and existing methods. Armed with an action tool library, the SymAgent, consisting of a planner and an executor, autonomously interacts with the KG environment to conduct reasoning.**

Existing work mainly adopts retrieval-augmented [6, 24, 26, 32] or semantic-parsing [1, 22, 43] methods to enhance the complex reasoning performance of LLMs with KG data. The former approaches rely on vector embeddings to retrieve and serialize the relevant

subgraph as input prompt for LLMs, while the latter employs LLMs to perform a structured search on KGs (e.g., SPARQL) to obtain answers. Despite their success, these methods share significant limitations. **Firstly**, they treat KGs merely as static knowledge repositories, overlooking the inherent reasoning patterns embedded in the symbolic structure of KGs. These patterns could substantially aid LLMs in decomposing complex problems and aligning the semantic granularity between natural language questions and KG elements. For instance, in Figure 1, given the question *Where was the person who recorded "I'm Gonna Get Drunk and Play Hank Williams" born?*, the symbolic rule $featured\_artist.recordings(e_1, e_2) \wedge person.place \_of\_birth(e_2, e_3)$ derived from the KG serves as an abstract representation of the question, revealing the intrinsic connection between question decomposition and KG structural patterns. In contrast, retrieval-based methods often suffer from superficial correlations, retrieving semantically similar but irrelevant information and even harmful disturbance, leading to degraded model performance. **Moreover**, both methods typically assume that all factual triples required for each question are entirely covered by the KG, which is unrealistic for manually curated KGs. When KGs fail to cover the necessary information, parser-based methods struggle to execute SPARQL queries effectively, limiting their ability to provide accurate answers or engage in complex reasoning tasks.

In light of these limitations, we delve into the exploration of the effective fusion of KGs and LLMs, enabling their collective augmentation in complex reasoning tasks. Fundamentally, realizing this integration poses several significant challenges: (i) **Semantic Gap**. Enabling the KG to participate deeply in the reasoning process of LLM requires aligning the symbolic structure of KGs with the neural representations of LLMs. (ii) **Incompleteness of KG**. When encountering insufficient information, it is necessary to retrieve relevant unstructured documents and identify missing triples consistent with the KG's semantic granularity during the reasoning process. (iii) **Learning with Limited Supervision**. The complexity of tasks and the current limitation of having only natural language input-output pairs make it difficult to unlock the full reasoning potential of LLMs.

To address these challenges, we propose SymAgent, a novel framework designed to autonomously and effectively integrate the capabilities of both LLM and KG. By treating the KG as a dynamic environment, we transform complex reasoning tasks into multi-step interactive processes, enabling in-depth analysis and proper decomposition of complex questions. Specifically, SymAgent comprises two key components: a planning module and an execution module. The planning module leverages LLM's inductive reasoning to derive symbolic rules from the KG, creating high-level plans for aligning natural language questions with the KG structure and employing it as a navigational tool. In the execution module, we extend the agent's capacity by curating a multi-functional toolbox, enabling the manipulation of both structured data and unstructured documents. By engaging in a thought-action-observation loop, the agent continuously reflects on the derived plan, action execution results, and past interactions to autonomously orchestrate action tools. This process not only allows for the collection of the necessary information to answer the question but also simultaneously identifies missing factual triples to complete the KG, addressing the challenge of KG incompleteness. Given the lack of well-annotated expert trajectories, we introduce a self-learning framework, which includes online exploration and offline iterative policy updates. Through continuous interaction with the KG environment, the agent can self-synthesize and refine trajectory data without human annotation, empowering performance improvement.

In summary, our main contributions are as follows:

- We propose SymAgent, a novel neural-symbolic driven LLM-based agent framework for complex reasoning over knowledge graphs, effectively integrating the strengths of both LLMs and KGs. SymAgent transforms natural language questions into multi-step interaction processes through the automatic invocation of pre-defined action tools, achieving mutual enhancement of KGs and LLMs.
- We develop an innovative self-learning framework involving iterative training of LLMs through interactions with the dynamic KG environment. The proposed framework eliminates the need for human annotation or a stronger teacher model, enabling autonomous self-improvement.
- Experimental results on several widely used complex reasoning datasets demonstrate that SymAgent with weak LLM backbones (i.e., 7B series) achieves better or comparable performance compared to strong baselines. Comprehensive empirical analyses validate the effectiveness of SymAgent in multiple aspects, including complex question decomposition, missing factual triples identification, and self-learning strategy.

## 2 RELATED WORKS

**Complex Reasoning over Knowledge Graph.** Complex Reasoning over Knowledge graph aims to provide answers to multi-hop natural language questions using knowledge graphs as their primary source of information [1, 18, 22]. Existing methods can be broadly categorized into semantic-parsing and retrieval-augmented methods. Semantic-parsing methods parse questions into the executable formal language (e.g., SPARQL) and perform precise queries on KGs to obtain answers [22, 43]. Initial works [2, 20] utilize strategies of step-wise query graph generation and search for parsing. Subsequent works [1] employ Seq2Seq models (e.g., T5 [28]) to generate SARSQL-expressions directly, which take advantage of the ability of pre-trained language models to enhance the semantic parsing process. More recently, ChatKBQA [22] further fine-tunes large language models (e.g., LLaMA [35]) to improve the accuracy of formal language generation. Despite these advancements, semantic-parsing methods heavily rely on the quality of generated queries, and no answers can be obtained if the query is not executable.

Retrieval-augmented methods [15, 16, 24] retrieve the relevant factual triples from the KG and then feed them to the LLM to help generate the final answers. Some methods [15] develop specialized interfaces for gathering pertinent evidence from structured data, while others [16, 26] retrieve facts by assessing semantic similarities between the question and associated facts. Meanwhile, certain approaches [5, 23] utilize the LLM to decompose the question and then retrieve corresponding triples for generation, enhancing the precision of the retrieval process. Notably, ToG [33] adopts an explore-and-exploit strategy, allowing the LLM to traverse the KG for information gathering, achieving state-of-the-art performance.

However, most of these approaches rely on capable closed-source LLM APIs (e.g., GPT4 [27]), resulting in significant performance degradation when using weak LLMs as backbones. Furthermore, they all assume that KGs comprehensively contain the answers, overlooking the issue of KG incompleteness in real-world scenarios.
**Large Language Model based Agents.** With the surprising long-horizon planning and reasoning capabilities shown in LLMs [13], researchers have explored building LLM-based agent systems [8] to unlock the door of Artificial General Intelligence. The most representative LLM agent, ReAct [41], proposes a prompting method to enable LLMs to interact with external environments and receive feedback. Subsequent works further focus on agent planning [31], function call [30], and code generation [4], improving the ability of LLMs on various complicated tasks. Recently, there has been an increasing focus on endowing open-source LLMs with agent capabilities through fine-tuning [38] on expert data distilled from teacher models. Furthermore, recent research emphasizes the significance of incorporating reinforcement learning techniques with LLMs to enhance decision-making in dynamic scenarios. Notably, studies like [7] highlight how RL frameworks can enable LLMs to continuously adapt their strategies with meticulously designed prompts, thus significantly improving their performance in practical applications. However, these approaches heavily rely on prompts for customization, which makes it difficult to tailor the behavior. In this paper, we introduce a self-learning framework, enabling weak LLMs to improve iteratively by interacting with the environment.

## 3 PRELIMINARIES

### 3.1 Symbolic Rules

A knowledge graph is a collection of factual triples, denoted as $\mathcal{G} = \{(e, r, e')|e, e' \in \mathcal{E}, r \in \mathcal{R}\}$, where $\mathcal{E}$ and $\mathcal{R}$ represent the sets of entities and relations, respectively. Symbolic rules in KGs are typically expressed as first-order logic formulae:

$$r_h(x, y) \leftarrow r_1(x, z_1) \wedge r_2(z_1, z_2) \wedge \ldots \wedge r_n(z_{n-1}, y), \quad (1)$$

where the left-hand side denotes the rule head with relation $r_h$ that can be induced by ($\leftarrow$) the right-hand rule body, the rule body forms a *closed chain*, with successive relations sharing intermediate variables (e.g., $z_i$), represented by the conjunction ($\wedge$) of body relations. A KG can be regarded as *groundings* of symbolic rules by substituting all variables $x, y, z$ with specific entities. For example, given the triples *(Sam, workFor, OpenAI), (OpenAI locatedIn SF)*, and *(Sam liveIn, SF)*, a grounding of the length-2 symbolic rule is $liveIn(Sam, SF) \leftarrow workFor(Sam, OpenAI) \wedge locatedIn(OpenAI, SF)$.

### 3.2 Task Formulation

In this paper, we transform the reasoning task on KG into an LLM-based agent task, where the KG serves as an environment providing execution feedback rather than merely acting as a knowledge base. The reasoning process can thus be viewed as a multi-step interaction with partial observations from the KG. This interactive process can be formalized as a Partially Observable Markov Decision Process (POMDP): $(\mathcal{Q}, \mathcal{S}, \mathcal{A}, \mathcal{O}, \mathcal{T})$ with question space $\mathcal{Q}$, state space $\mathcal{S}$, action space $\mathcal{A}$, observation space $\mathcal{O}$, and state transition function $\mathcal{T} : \mathcal{S} \times \mathcal{A} \rightarrow \mathcal{S}$. Note that in our language agent scenario, $\mathcal{Q}$,

$\mathcal{A}$, and $\mathcal{O}$ are subspaces of the natural language space, and the transition function $\mathcal{T}$ is determined by the environment.

Given a question $q \in \mathcal{Q}$ and the KG $\mathcal{G}$, the LLM agent generates the action $a_0 \sim \pi_\theta(\cdot|q, \mathcal{G}) \in \mathcal{A}$ based on its policy $\pi_\theta$. This action leads to a state transition, and the agent receives execution feedback as observation $o_0 \in \mathcal{O}$. The agent then continues to explore the environment until an appropriate answer is found or another stop condition is met. The historical trajectory $\mathcal{H}_n$ at step $n$, consisting of a sequence of actions and observations, can be represented as:

$$\mathcal{H}_n = (q, \mathcal{G}, a_0, o_o, \ldots, a_{n-1}, o_{n-1}) \sim \pi_\theta(\mathcal{H}_n|q, \mathcal{G}),$$

$$\pi_\theta(\mathcal{H}_n|q, \mathcal{G}) = \prod_{j=1}^{n} \pi_\theta(a_j|q, \mathcal{G}, a_0, o_0, \ldots, o_{j-1}), \quad (2)$$

where $n$ is the total interaction steps. Finally, the final reward $r(q, \mathcal{H}_n) \in [0, 1]$ is computed, with 1 indicating a correct answer.

## 4 METHODOLOGY

In this section, we present the proposed SymAgent, which effectively synergizes the cognitive potential inherent in knowledge graphs (KGs) with the reasoning and information integration capabilities of LLMs to autonomously tackle complex reasoning tasks over KGs. SymAgent consists of an Agent-Planner and an Agent-Executor. The Agent-Planner derives symbolic rules embedded within the KG to decompose the question and orchestrate the reasoning steps (Section 4.1). Building upon these symbolic rules, the Agent-Executor invokes actions to answer the question by synthesizing insights derived from agent reflection and observed environment feedback (Section 4.2). Given the lack of well-annotated step-by-step reasoning data, we further introduce a self-learning framework that facilitates SymAgent and the KG to augment collaboratively through autonomous interaction with the environment (Section 4.3). The overall architecture of SymAgent is illustrated in Figure 2.

### 4.1 Agent-Planner Module

The Agent-Planner functions as a high-level planner, leveraging LLM's reasoning capability to decompose questions into executable reasoning chains. However, we observed that merely prompting the LLM to plan the entire reasoning workflow does not yield satisfactory performance. Current LLMs struggle to align complex questions with the semantics and connectivity patterns of the KG, resulting in coarse-grained reasoning chains that are ineffective for precise information retrieval and integration.

To address this limitation, we employ the LLM to identify potential symbolic rules within the KG that could answer the question rather than generating detailed step-by-step plans. On the one hand, LLMs have been demonstrated to be effective inductive reasoners but poor deductive reasoners [48]. On the other hand, symbolic rules inherently reflect the reasoning patterns of KG, serving as implicit information to aid in decomposing complex questions. In this way, the Agent-Planner establishes a bridge between natural language questions and structural information of KG, enhancing both the accuracy and generalizability of the reasoning process.

Specifically, given a question $q$, we employ BM25 [29] to retrieve a set of seed questions $\{q_{seed_i}\}_{i=1}^{k}$ from the training set, where each $q_{seed_i}$ shares similar question structure with $q$, potentially

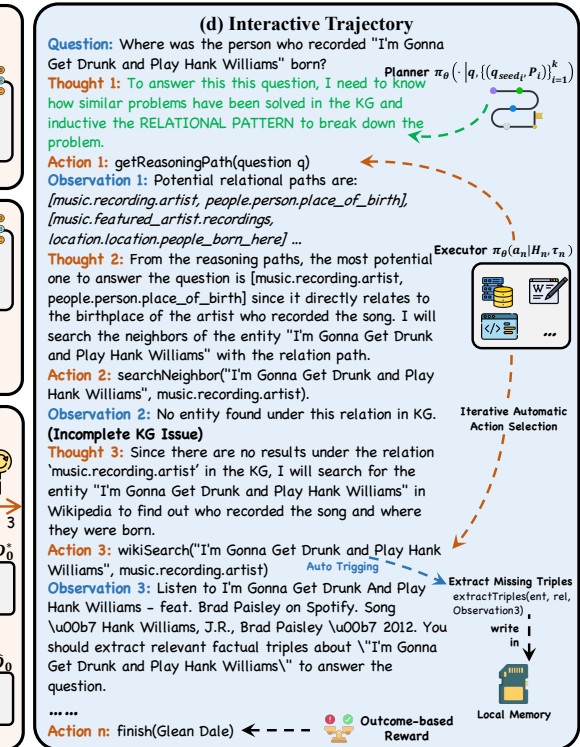

**Figure 2: The overview of our proposed SymAgent. (a) the planner in SymAgent, which derives the symbolic rules from the KG to guide the reasoning; (b) the executor in SymAgent, which conducts the automatic action invocation to obtain the answer; (c) the self-learning framework to enhance the agent iteratively; and (d) an example of the synthesized action invoking trajectory.**

requiring analogous solution strategies. For each $q_{seed_i}$, we adopt breadth-first-search (BFS) to sample a set of closed paths $P_i = \{p_{i_1}, p_{i_2}, \ldots, p_{i_m}\}$ from the query entity $e_q$ to the answer entity $e_a$ within the KG $\mathcal{G}$, where $p_{i_j} = r_1(e_q, e_1) \wedge r_2(e_1, e_2) \ldots \wedge r_L(e_{L-1}, e_a)$ is a sequence of relations. These closed paths can be considered as groundings of symbolic rules that answer the question. We then generalize these closed paths by replacing specific entities with variables, transforming them into rule bodies shown in Equation 1. This process constructs few-shot demonstrations $\mathcal{M} = \{(q_{seed_i}, P_i)\}_{i=1}^k$ to prompt our SymAgent to generate appropriate rule bodies for $q$:

$$p \sim \pi_\theta(\cdot | \rho_{Plan}, q, \mathcal{M}), \qquad (3)$$

where $\rho_{Plan}$ stands for the prompt to instruct the rule bodies generation. The generated KG-aligned symbolic rules $p$ serve to guide SymAgent's global planning and prevent it from falling into blind trial-and-error during the reasoning process.

## 4.2 Agent-Executor Module

Building upon the generated symbolic rules from KG $\mathcal{G}$, the Agent-Executor engages in a cyclical paradigm of *observation*, *thought*, and *action* to navigate the autonomous reasoning process. In contrast to existing methods that retrieve information from the KG, potentially introducing large amounts of irrelevant data, the Agent-Executor leverages expert feedback from the KG structure to dynamically adjust the reasoning process. This approach enables KGs, which

store a wealth of informative and symbolic facts, to deeply participate in the reasoning process together with LLMs rather than being merely treated as a static repository of information.

*4.2.1 Action Space.* Given that LLMs cannot directly process the structured data in KGs, and considering the need to rely on external unstructured documents during the reasoning process to address the issue of incomplete information in KGs, we define the agent's action space as a set of functional tools. By leveraging the function call capabilities of LLMs, our SymAgent not only overcomes the limitations of LLMs in handling structured data but also provides a flexible mechanism for integrating diverse information sources, thereby enhancing the agent's reasoning capabilities and adaptability. The action space consists of the following functional tools:

- *getRules(sub_question)*: receives the sub_question as input and returns potential symbolic rules. As depicted in Equation 3, this action leverages the inductive reasoning capability of LLMs to generate KG-aligned symbolic rules that decompose the sub_question, effectively guiding the reasoning process.
- *searchWikidata(ent, rel)*: retrieves relevant documents from Wikipedia when KG information is insufficient. This action bridges structured KG data with unstructured text, enhancing reasoning with incomplete information.

- $extractTriples(ent, rel, doc)$: extracts triples related to the current query's entity and relation from retrieved documents. Notably, this action is not explicitly invoked by the agent but automatically triggered after $searchWikidata$ is called. The extracted triples are aligned with the KG's semantic granularity and can be integrated into the KG, facilitating its expansion.
- $searchNeighbor(ent, rel)$: is a graph exploration function. It returns neighbors of a particular entity under a given relation in the KG, enabling efficient traversal and discovery of related entities.
- $finish(e_1, e_2, \ldots, e_n)$ returns a list of answer entities, indicating that the final answers have been obtained and the reasoning process has concluded.

*4.2.2 Interactive Process.* Treating the KG as the environment and the results of action executions as observations, the entire reasoning process becomes a sequence of agent action calls and corresponding observations. We adopt a react-style approach [41], which generates a chain-of-thought rationale before taking actions, reflecting on the current state of the environment. Formally, we extend the Equation 2, and the interaction trajectory at step $n$ can be further represented as:

$$\mathcal{H}_n = (q, \mathcal{G}, p, \tau_0, a_0, o_0, \ldots, \tau_{n-1}, a_{n-1}, o_{n-1}), \quad (4)$$

where $\tau$ is the internal thought of the agent by reflecting on the historical trajectory, $a$ is an action selected from the tool set defined above, and $o$ is the observation determined by executing an action. Based on this historical trajectory, the process for generating the subsequent thought $\tau_n$ and action $a_n$ can be formulated as:

$$\pi_\theta(\tau_n | \mathcal{H}_n) = \prod_{i=1}^{|\tau_n|} \pi_\theta(\tau_n^i | \mathcal{H}_n, \tau_n^{<i}),$$

$$\pi_\theta(a_n | \mathcal{H}_n, \tau_n) = \prod_{j=1}^{|a_n|} \pi_\theta(a_n^j | \mathcal{H}_n, \tau_n, a_n^{<j}), \quad (5)$$

where $\tau_n^i$ and $|\tau_n|$ represent the $i$-th token and the total length of $\tau_n$, $a_n^j$ and $|a_n|$ represent the $j$-th token and the total length of $a_n$. The agent loop continues until either the $finish()$ action is invoked or it reaches the predefined maximum iterative steps.

## 4.3 Self-learning

Given that the initial dataset comprises only question-answer pairs without well-annotated step-by-step interaction data, we propose a self-learning framework. Rather than distilling reasoning chains from more capable models (e.g., GPT-4 [27]), our approach enables weak policy LLM $\pi_\theta$ to interact with the environment adequately, thereby improving through self-training. The self-learning process consists of two primary phases: online exploration and offline iterative policy updating.

*4.3.1 Online Exploration.* In this phase, the base agent $\pi_{\theta_0}$ interacts with the environment autonomously through a thought-action-observation loop according to Section 4.2.2, synthesizing a set of initial trajectories $\mathcal{U}_0 = \{\mu_1, \mu_2, \ldots, \mu_N\}$. For each trajectory $\mu_i$, we employ an outcome-based reward mechanism, defining the reward

as the final answer's recall value:

$$r(\mu_i) = \text{Recall}(A_{\mu_i}, A_{gt}) = \frac{|A_{\mu_i} \cap A_{gt}|}{|A_{gt}|}, \quad (6)$$

where $A_{\mu_i}$ is the set of answer entities extracted from the final action of trajectory $\mu_i$, and $A_{gt}$ is the set of ground truth answer entities. This process yields a collection of self-explored trajectories $\mathcal{D}_0 = \{(\mu_i, r(\mu_i))\}_{i=1}^N$.

To address the potential errors in agent action invocation (e.g., incorrect tool invocation formats) that may impair exploration effectiveness, we leverage the LLM's self-reflection capability to refine the trajectories. Using $\mathcal{D}_0$ as reference, the policy LLM $\pi_{\theta_0}$ regenerate new refined trajectories, formulated as $\{\hat{\mu}_i\}_{i=1}^N \sim \pi_{\theta_0}(\cdot | \mu_i, r(\mu_i))$. After applying the same reward mechanism, we can get a refined trajectory collection $\widehat{\mathcal{D}_0} = \{(\hat{\mu}_i, r(\hat{\mu}_i))\}_{i=1}^N$.

After self-exploration and self-reflection, we obtain two trajectory collections of equal size: $\mathcal{D}_0$ and $\widehat{\mathcal{D}_0}$. To enhance the quality of candidate trajectories, we employ a heuristic method to merge these two collections, resulting in an optimized trajectory set. Following the principle of final answer consistency, we obtain the merged trajectory collection $\mathcal{D}_0^* = \{(\mu_i^*, r(\mu_i^*))\}_{i=1}^N$:

$$\mathcal{D}_0^*(i) = \begin{cases} (\mu_i, r(\mu_i)), & \text{if } r(\mu_i) > r(\hat{\mu}_i), \\ (\hat{\mu}_i, r(\hat{\mu}_i)), & \text{if } r(\mu_i) < r(\hat{\mu}_i), \\ (t, r(t)), & \text{if } r(\mu_i) = r(\hat{\mu}_i) > 0, \\ \text{filtered}, & \text{if } r(\mu_i) = r(\hat{\mu}_i) = 0. \end{cases} \quad (7)$$

In this equation, $t = \arg\min_{s \in \{\mu_i, \hat{\mu}_i\}} |s|$ denotes that we select the trajectory with the shorter length when the rewards are equal and non-zero.

*4.3.2 Offline Iterative Policy Updating.* Given the merged trajectories $\mathcal{D}_0^*$, an intuitive way to improve the performance of the agent is fine-tuning with these trajectories. Under an auto-regressive manner, the loss of the agent model can be formulated as:

$$\mathcal{L}_{SFT} = -\mathbb{E}_{\mu \sim \mathcal{D}^*}[\pi_\theta(\mu | q)],$$

$$\pi_\theta(\mu | q) = -\sum_{j=1}^{|\mathcal{X}|} (\mathbb{1}(x_j \in \mathcal{A}) \times \log \pi_\theta(x_j | q, x_{<j})), \quad (8)$$

where $\mathbb{1}(x_j \in \mathcal{A})$ is the indicator function about whether $x_j$ is a token belonging to thoughts or actions generated by the agent.

After updating the policy model parameters , we employ an iterative optimization approach to continuously improve the performance of agent. The updated model undergoes repeated cycles of self-exploration, self-reflection, and trajectory merging on the initial dataset, generating new trajectory data for further fine-tuning. This iterative process continues until the improvement in performance on the validation set becomes negligible, at which point we terminate the iteration.

## 5 EXPERIMENTS

In this section, we evaluate SymAgent on widely used datasets. We conduct extensive experiments to show the effectiveness of our method by answering the following research questions:

- **RQ1:** How does SymAgent perform compared to state-of-the-art (SOTA) baselines across various complex reasoning datasets?

**Table 1: Results (%) of SymAgent. The best results are marked in bold, and the second-best results are marked with underline. ◗ denotes the prompt-based, while ● denotes the fine-tuned methods. ∗ denotes the unseen dataset (i.e., zero-shot setting).**

| Backbone | Method | WebQSP | | | CWQ | | | MetaQA-3hop* | | |
|---|---|---|---|---|---|---|---|---|---|---|
| | | Hits@1 | Accuracy | F1 | Hits@1 | Accuracy | F1 | Hits@1 | Accuracy | F1 |
| **GPT-4** | ◗ CoT [36] | 56.68 | 46.97 | 39.98 | 41.46 | 37.05 | 35.74 | 43.50 | **26.48** | 22.86 |
| | ◗ w.t. Retrieval | 53.03 | 43.57 | 38.95 | 38.92 | 36.23 | 33.77 | 32.00 | 11.29 | 12.61 |
| **LLaMA2-7B** | ◗ CoT [36] | 38.87 | 29.95 | 25.02 | 13.29 | 12.02 | 11.45 | 2.00 | 0.81 | 0.85 |
| | ◗ RaAct [41] | 30.36 | 19.86 | 18.91 | 12.66 | 11.12 | 10.57 | 15.00 | 5.98 | 6.65 |
| | ◗ ToG [33] | 29.15 | 20.51 | 19.39 | 15.19 | 13.96 | 13.24 | 14.00 | 4.75 | 5.13 |
| | ● RoG [24] | 47.77 | 30.07 | 31.39 | 27.53 | 25.28 | 24.68 | 21.00 | 8.73 | 8.27 |
| | ● **SymAgent (Ours)** | **55.47** | **39.73** | **41.27** | **35.13** | **30.45** | **31.15** | **37.00** | **15.47** | **16.87** |
| **Mistral-7B** | ◗ CoT [36] | 34.82 | 26.96 | 23.41 | 26.90 | 24.20 | 22.81 | 9.00 | 2.07 | 2.63 |
| | ◗ RaAct [41] | 29.55 | 20.61 | 20.34 | 19.94 | 16.82 | 15.88 | 16.00 | 8.42 | 7.65 |
| | ◗ ToG [33] | 31.98 | 21.69 | 22.11 | 20.57 | 18.19 | 16.65 | 16.50 | 8.29 | 7.77 |
| | ● RoG [24] | 50.61 | 32.91 | 34.22 | 28.16 | 25.92 | 25.31 | 29.50 | 12.13 | 12.92 |
| | ● **SymAgent (Ours)** | **61.94** | **45.02** | **47.08** | **37.66** | **33.51** | **34.05** | **38.50** | **14.82** | **16.12** |
| **Qwen2-7B** | ◗ CoT [36] | 27.13 | 20.19 | 19.15 | 29.11 | 24.90 | 24.68 | 6.50 | 2.78 | 3.41 |
| | ◗ ReAct [41] | 40.49 | 29.10 | 28.90 | 25.32 | 22.36 | 22.12 | 20.50 | 7.98 | 7.72 |
| | ◗ ToG [33] | 54.25 | 38.16 | 39.28 | 30.70 | 27.09 | 26.76 | 26.50 | 10.09 | 9.73 |
| | ● RoG [24] | 51.82 | 34.12 | 35.44 | 29.11 | 26.87 | 26.26 | 25.00 | 11.86 | 11.92 |
| | ● **SymAgent (Ours)** | **78.54** | **57.48** | **57.05** | **58.86** | **50.19** | **48.30** | **57.00** | 24.68 | **25.76** |

- **RQ2:** What is the contribution of each key module in our SymAgent framework to the overall performance?
- **RQ3:** How effective is the proposed self-learning framework compared to distillation from teacher models?
- **RQ4:** To what extent can SymAgent enhance KGs by identifying missing triples and facilitating automatic KG completion?

| Dataset | Train | Valid | Test | Max Hop | $|\mathcal{G}|$ |
|---|---|---|---|---|---|
| WebQSP | 2,826 | 120 | 247 | 2 | 8,309,195 |
| CWQ | 1,635 | 120 | 316 | 4 | 8,309,195 |
| MetaQA-3hop | - | - | 200 | 3 | 133,582 |

**Table 2: Statistics of the datasets. $|\mathcal{G}|$ denotes the number of triples in the background KG for each dataset.**

## 5.1 Experimental Setup

*5.1.1 Datasets.* We adopt three popular knowledge graph question answer datasets: WebQuestionSP (WebQSP) [42], Complex WebQuestions (CWQ) [34], and MetaQA-3hop [10] for evaluation. WebQSP and CWQ datasets are constructed from commonsense KG Freebase [3], which contain up to 4-hop questions. MetaQA-3hop is based on a domain-specific movie KG, and we specifically select this dataset to evaluate the zero-shot reasoning performance of our model in a specific domain scenario. This means we only train on CWQ and WebQSP and then perform in-context reasoning on MetaQA-3hop to assess the model's generalization capabilities to unseen relation types. To further simulate incomplete KGs, we adopt a breadth-first search method to extract paths from the question entity to the answer entity and then randomly remove some triples. In this scenario, semantic parsing methods fail to obtain the correct answers due to unexecutable formal expressions. The detailed construction process can refer to Appendix A.1. To better evaluate model performance on complex reasoning tasks, we sample a subset from the test sets that specifically require multi-hop reasoning to solve the questions. The statistics of the resulting datasets are presented in Table 2.

*5.1.2 Baselines.* We evaluate the performance of SymAgent with three different LLM backbones: (i) Mistral-7B [14] (Mistral-7B-Instruct-v0.2 version), (ii) LLaMA2-7B [35] (Meta-LLaMA-2-7B-Chat version), and (iii) Qwen2-7B [39] (Qwen2-7B-Instruct version). Our method is compared against two prompt-based baselines: CoT [36] and ReAct [41]. Additionally, we include two strong baselines, ToG [33] and RoG [24]. ToG employs an explore-and-exploit strategy, while RoG adopts a retrieval-augmented approach, effectively coupling KG and LLM to achieve state-of-the-art performance. Notably, we have not included semantic parsing methods in our comparisons. This is because, in the incomplete KG scenario, the formal expressions generated by these methods are often unexecutable, rendering them ineffective for this task. To provide a comprehensive evaluation, we also incorporate comparisons with GPT-4 (gpt-4-32K-0613) using document retrieval augmentation. All prompt-based baselines are tested under one-shot settings, while the fine-tuning-based baselines are trained using LoRA [12]. For detailed prompts used in our experiments, please refer to Appendix A.3. Following the previous setting, we adopt Accuracy, Hits@1,

and F1 scores as metrics. The implementation detail can refer to Appendix A.2.

## 5.2 Performance Comparison with SOTA (RQ1)

The experimental performance of our SymAgent compared to SOTA methods is presented in Table 1. The overall results demonstrate that SymAgent consistently achieves superior performance across all datasets, validating the effectiveness of our approach.

First, SymAgent demonstrates consistent improvement across all LLM backbones compared to both prompt-based and fine-tuned methods, which underscores SymAgent's adaptability and robustness. In particular, SymAgent, with Qwen2-7B backbone, achieves the best performance, outperforming GPT-4 across all three datasets with average improvements of 37.19% in Hits@1, 16.87% in Accuracy, and 30.17% in F1 score. The superior performance can be attributed to the better function-calling capabilities of Qwen2 compared to the other two backbones, which often encounter action tool calling errors (e.g., extra arguments). This demonstrates that our method can effectively leverage the strengths of more advanced LLM, enhancing the overall performance in complex reasoning tasks.

Moreover, GPT-4 performance between CoT and retrieval augmented reveals that direct document retrieval for complex questions can harm performance, especially in domain-specific tasks. For instance, in MetaQA-3hop, the F1 score degrades by 10.25 (from 22.86 to 12.61) when using retrieval augmentation. The potential reason is that shallow vector retrieval introduces semantically similar but irrelevant noisy information [45]. A similar trend is observed with weaker LLMs. Interestingly, when the base model has adequate instruction-following capabilities (e.g., Qwen2-7B), ToG outperforms the fine-tuned RoG. The potential reason is that the explore-and-exploit strategy can leverage the LLM's inherent knowledge to address the incompleteness issue of KG, whereas RoG relies heavily on path retrieval and struggles in such a scenario. In contrast, our SymAgent can fully utilize the advantages of both KG and LLM, effectively decomposing problems and achieving excellent performance.

Finally, by comparing the performance of SymAgent across different datasets, we observe that SymAgent shows a larger improvement ratio on the more challenging CWQ dataset, demonstrating its capability to handle complex reasoning problems. Furthermore, from the results on MetaQA-3hop, we can observe that LLMs lacking domain knowledge perform worse, while our SymAgent can significantly enhance the backbone's capabilities. This improvement is particularly notable in the zero-shot setting, where SymAgent achieves a remarkable 6× increase in F1 score compared to the base LLM, highlighting its ability to generalize and reason effectively in unseen scenarios. In the following ablation and further analysis experiments, unless otherwise specified, we adopt Qwen2-7B as the backbone of SymAgent due to its superior performance.

## 5.3 Ablation Study (RQ2)

In this section, we conduct a series of ablation experiments to analyze the contribution of each component in SymAgent. To validate the planner module (PM), executor module (EM), and self-learning framework (SL), we systematically remove these components to

| | PM | EM | SL | WebQSP | | CWQ | |
|---|---|---|---|---|---|---|---|
| | | | | Hits@1 | F1 | Hits@1 | F1 |
| **Variants** | ✔ | - | - | 50.61 | 34.22 | 29.43 | 26.34 |
| | - | ✔ | - | 55.06 | 40.58 | 33.54 | 28.95 |
| | ✔ | ✔ | - | 64.37 | 47.48 | 37.66 | 32.65 |
| | - | ✔ | ✔ | 56.68 | 41.73 | 38.92 | 32.63 |
| **SymAgent** | ✔ | ✔ | ✔ | 78.54 | 57.05 | 58.86 | 48.30 |

**Table 3: Ablation study for the SymAgent with Qwen2-7B.**

create variants for comparison. Ablation results in Table 3 reveal that all components are essential because their absence has a detrimental effect on performance. Specifically, we argue that deriving symbolic rules from the KG is vital, which can be demonstrated by comparing the experimental results between SymAgent and EM + SL, as well as PM + EM and EM-only. Similarly, by comparing PM-only and PM+EM, we can find that arming the model with action tools to access unstructured documents and structured KGs achieves notable improvement. Moreover, by comparing the results between EM-only and EM+SL, we can find self-learning makes minor improvements, the potential low quality of the self-synthesized trajectories without a planner module. Overall, these findings demonstrate the effectiveness of our modular approach, with each component contributing uniquely to SymAgent in handling complex reasoning tasks.

## 5.4 Analysis on Self-learning Framework (RQ3)

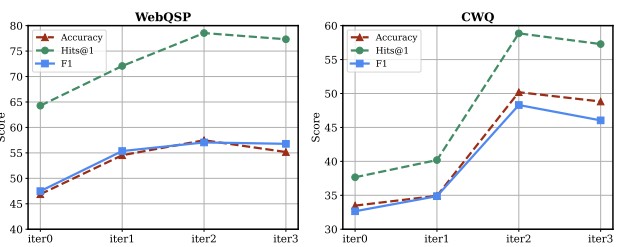

**Figure 3: The impact of the iteration numbers in the self-learning phase on model performance.**

***The Number of Iterations.*** Figure 3 presents a comparative analysis of the effects of the number of iterations during the self-learning phase. In the initial stages of iterative training, we observe a rapid improvement in model performance, validating the effectiveness of our self-refine and heuristic merge methods in acquiring substantial trajectory data. This iterative approach enables the model to thoroughly explore the environment, thereby enhancing its performance. Consistent with previous work [17], these findings corroborate the efficacy of iterative training under rejection sampling in bolstering the model's comprehension of the training data. However, as the number of iterations increases, we notice fluctuations in model performance. This phenomenon can be attributed to our use of outcome-based rewards. In practice, the model may produce

correct final results despite errors in intermediate steps. Continued iteration with these trajectories can lead to the model fitting these spurious correlations. This observation highlights the need for more nuanced evaluation metrics and reward mechanisms in future iterations of the self-learning framework.

***Roles of Self-refinement & Heuristic Merging.*** To further explore the roles of self-refinement and heuristic merging within our self-learning framework, we designed two variant training recipes: 1) $-self\text{-}refine$, which solely employs rejection sampling for trajectory data acquisition, and 2) $-merge$, which directly utilizes refined trajectories as the training set without merging. The experimental results, as presented in Table 4, demonstrate that the removal of either component adversely affects the model's performance. The full self-learning model consistently outperforms its variants across all metrics on both WebQSP and CWQ datasets. On WebQSP, removing self-refinement decreases Hits@1 by 2.43 percentage points, while removing merging leads to a 1.62 percentage point drop. Similar trends are observed for Accuracy and F1 scores, as well as on the CWQ dataset. These findings highlight the synergistic effect of self-refinement and heuristic merging in our framework. Self-refinement likely increases trajectory quantity, while merging further enhances quality.

| | WebQSP | | | CWQ | | |
|---|---|---|---|---|---|---|
| | Hits@1 | Accuracy | F1 | Hits@1 | Accuracy | F1 |
| **Self Learning** | **78.54** | **57.48** | **57.05** | **58.86** | **50.19** | **48.30** |
| $-self\text{-}refine$ | $76.11_{\downarrow 2.43}$ | $56.22_{\downarrow 1.26}$ | $56.50_{\downarrow 0.55}$ | $56.33_{\downarrow 2.53}$ | $48.75_{\downarrow 1.44}$ | $46.08_{\downarrow 2.22}$ |
| $-merge$ | $76.92_{\downarrow 1.62}$ | $55.7_{\downarrow 1.78}$ | $55.62_{\downarrow 1.43}$ | $57.28_{\downarrow 1.58}$ | $48.83_{\downarrow 1.36}$ | $46.04_{\downarrow 2.26}$ |
| **Distilling** | $77.32_{\downarrow 1.22}$ | $55.17_{\downarrow 2.31}$ | $56.78_{\downarrow 0.27}$ | $54.43_{\downarrow 4.43}$ | $46.18_{\downarrow 4.01}$ | $42.98_{\downarrow 5.32}$ |

**Table 4: The impact of different training recipes on model performance. And the comparison between distilling from the teacher model and the self-learning framework.**

***Distilled Trajectories v.s. Self-synthesized Trajectories.*** We adopt a conventional data synthesis approach to generate trajectory data from a capable teacher model (GPT-4) and use these data to fine-tune our model. The experimental results are presented in the 4-$th$ row of Table 4. We observe that while the distilling approach shows competitive performance, it consistently underperforms our self-learning framework across all datasets. The performance gap is more pronounced on the CWQ dataset, with decreases of 4.43, 4.01, and 5.32 percentage points in Hits@1, Accuracy, and F1 scores, respectively. This is because responses from a similar model are *easier-to-fit* than those from a more capable model, resulting in reduced memorization [17]. Considering the extremely high costs and cumbersome prompt optimizations, this training approach is far from sustainable compared to a self-learning framework.

## 5.5 Quality of Extracted Triples & Error Type Analysis (RQ4)

***Quality of Extracted Triples.*** Armed with a comprehensive action tool set, our SymAgent addresses KG incompleteness by leveraging both structured and unstructured data. The WikiSearch action triggers an extract action to identify missing triples from retrieved texts, effectively aligning and enriching the KG with external information. To validate this approach, we augment the KG with

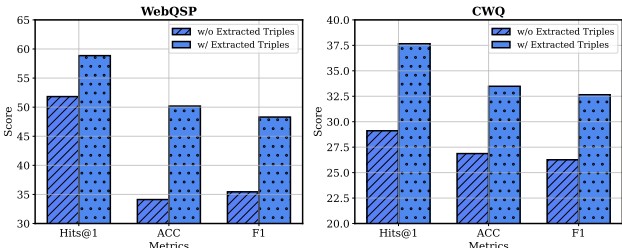

**Figure 4: Performance of RoG on KG augmented with triples extracted by our model.**

SymAgent-identified triples and test a retrieval-augmented generation model RoG on this enhanced KG. As shown in Figure 4, the results demonstrated a significant improvement in the performance of RoG, providing empirical evidence that the quality of triples identified by our method is sufficient for integration into existing KGs. This finding not only validates our approach but also suggests a potential synergistic enhancement between LLM and KG through our SymAgent.

**Error Analysis.** To gain deeper insights into our model's performance, we conducted an error analysis by categorizing the failure cases into four types: 1) Invalid Action (IA), where the model invokes an action not defined in the action tool set, 2) Error in Arguments (EA), where insufficient or excessive arguments are provided, 3) Exceeding Maximum Steps (EMS), where the reasoning steps exceed the predefined maximum number of steps, and 4) Reasoning Error (RE), where the final answer is incorrect despite valid actions and steps. Table 5 presents the distribution of these error types across WebQSP, CWQ, and MetaQA-3hop datasets. WebQSP errors are predominantly RE (94.34%), while CWQ and MetaQA-3hop show more diverse distributions with significant EMS errors, indicating potential areas for targeted improvements in the future.

| Dataset | IA | EA | EMS | RE |
|---|---|---|---|---|
| WebQSP | 3.77 | 0.0 | 1.89 | 94.34 |
| CWQ | 2.31 | 10.00 | 23.08 | 64.61 |
| MetaQA-3hop | 3.49 | 12.79 | 39.53 | 44.19 |

**Table 5: Proportions (%) of different error types.**

## 6 CONCLUSION

In this paper, we introduce SymAgent, an automatic agent framework that synergizes LLM with structured knowledge to conduct complex reasoning over KG. Our method involves utilizing symbolic rules in KG to guide question decomposition, automatically invoking action tools to address the incompleteness issue of KG, and employing a self-learning framework for trajectory synthesis and continuous improvement. This multifaceted approach not only enhances the planning abilities of the agent but also proves effective in complex reasoning scenarios. Extensive experiments demonstrate the superiority of SymAgent, showcasing the potential to foster mutual enhancement between KG and LLM.

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

# A APPENDIX

## A.1 Construction Process of Dataset

In Algorithm 1, we demonstrate the detailed process of obtaining potential triples of a question and construct corresponding datasets. We employ BFS to get the reasoning path in KG and drop triples randomly to mimic the incomplete scenario.

| Hyperparameters | Value |
|---|---|
| *Training* | |
| lora_r | 32 |
| lora_alpha | 32 |
| lora_dropout | 0.05 |
| lora_target_modules | {q, k, v, o, down, up, gate}_proj |
| per_device_batch_size | 2 |
| gradient_accumulation_steps | 2 |
| warmup_ratio | 0.05 |
| self-learning iterations | 2 |
| *Inference* | |
| temperature | 0.1 |
| top_p | 0.9 |
| top_k | 600 |
| max_new_tokens | 512 |
| max_infer_step | 10 |

**Table 6: Detail hyperparameters used in our method.**

Prompt Template of Our SymAgent

```
You are a knowledge graph (KG) question-answering agent that interacts
with a KG storing factual knowledge. When a user asks a question, solve it
using interleaving Thought, Action, and Observation steps. Follow this
strict format: "Thought: your thoughts.\nAction: your next action."
Available actions:
1. getReasoningPath(entity, relation): Retrieve relational reasoning paths
to answer the question. Use this first as a high-level planning step.
2. searchNeighbor(entity, relation): Search the neighbors of the entity
with the specified relation in the KG.
3. wikiSearch(entity, relation): Search Wikipedia for the entity with
respect to the relation if the KG returns no relevant results. Extract
relevant triples (entity, relation, entity) from the Wikipedia page.
4. finish(entity1, entity2, ..., entityN): Conclude the conversation with
the final answer(s).

Steps to follow:
Start with getReasoningPath to obtain potential relational reasoning paths.
Follow the most plausible path step-by-step using searchNeighbor for each
relation in the path (e.g., for path r1 -> r2, first use searchNeighbor(e1,
r1), then use searchNeighbor(e2, r2)).
If searchNeighbor returns no valid information, use wikiSearch and extract
relevant triples.
Continue following the relational reasoning path until enough information
is gathered to answer the question.
Use finish to provide the final answer(s).
Use the following response format:
Thought: <your thoughts>
Action: <your next action>
Here are some examples:
{examples}
(END OF EXAMPLES)
```

**Figure 5: Prompt template of our SymAgent.**

Prompt Template of Action Extract

```
Here is the document about the entity {ent}:
{document}. You should extract relevant factual
triples about {ent} under the relation {rel},
which are beneficial to answer the question
{question}. You should only output the triples in
the form of [[entity, relation, object], ...]
```

**Figure 6: The extraction prompt for extraction action.**

Case Study

```
Question: who did viggo mortensen play in lord of the rings?
Thought 1: To answer this question, I need to know the surrounding
relational patterns of the entity Viggo Mortensen.
Action 1: getReasoningPath(Viggo Mortensen, played in).
Observation 1: Surrounding relational reasoning paths are:
[fictional_universe.work_of_fiction.part_of_these_fictional_universes,
fictional_universe.fictional_character.appears_in_these_fictional_univers
es], [film.film.starring, film.performance.character],
[media_common.adapted_work.adaptations, book.book.characters],
[film.film_series.films_in_series, book.book.characters],
[media_common.adaptation.adapted_from,
book.book_character.appears_in_book], [film.performance.actor,
film.performance.character], [film.actor.film,
film.performance.character], [people.person.gender,
fictional_universe.fictional_character.gender], [film.performance.film,
film.performance.character], [music.featured_artist.recordings,
book.book_character.appears_in_book], [film.film.story_by,
fictional_universe.fictional_character.character_created_by],
[film.actor.film, film.film_character.portrayed_in_films]. There are
multiple paths, maybe you should select one most potential path.
Thought 2: From the reasoning paths, the most potential one to answer the
question is [film.film.starring, film.performance.character] since it
relates to the actor playing a character in which he starred in. I will
search the neighbors of the entity Viggo Mortensen with the relation path.
Action 2: searchNeighbor(Viggo Mortensen, film.film.starring).
Observation 2: No entity found under this relation in the knowlege graph.
You can keep searching on graph or go to invoke wikiSearch to retrieve
relevant documents. You can also just answer if you think you can based
on the information available and your knowledge.
Thought 3: Since there are no results under the relation
'film.film.starring' in the knowledge graph, I will search for the entity
Viggo Mortensen in Wikipedia to find out the role he played in the Lord
of the Rings series.
Action 3: wikiSearch(Viggo Mortensen, film.film.starring).
Observation 3: By searching, Viggo Mortensen's relevant documents are
Known for ; The Lord of the Rings: The Return of the King. 9.0. Aragorn ;
Captain Fantastic. 7.8. Ben ; The Lord of the Rings: The Two Towers. 8.8.
Aragorn. He went on to play supporting roles in several notable films
during the 1990s, including The Indian Runner (1991), Carlito's Way
(1993), Crimson Tide (1995),.. You should extract relevant factual
triples about Viggo Mortensen to answer the question.
Thought 4: Based on the information from Wikipedia, I can get relevant
information in the triples of (Viggo Mortensen, film.film.starring, The
Lord of the Rings: The Return of the King), (Viggo Mortensen,
film.film.starring, The Lord of the Rings: The Two Towers), and (Viggo
Mortensen, film.film.starring, Captain Fantastic). According to the
collected information, the next step is to get the role he played in the
Lord of the Rings series according to the reasoning path
film.performance.character to get the role.
Action 4: searchNeighbor(The Lord of the Rings: The Return of the King,
film.performance.character).
Observation 4: Aragorn.
Thought 5: According to the reasoning path, I have found the character
Viggo Mortensen played in the Lord of the Rings series. I can conclude
the conversation.
Action 5: finish(Aragorn).
```

**Figure 7: A representative trajectory of SymAgent during complex reasoning.**

---

**Algorithm 1** The detailed process of dataset construction.

---

**Require:** Question $q$, Question entity $q\_ent$, Answer entity list $a\_ent\_list$, Knowledge Graph $\mathcal{G}$

**Ensure:** Final graph $\mathcal{G}'$

1: Initialize $L \leftarrow [], \mathcal{G}' \leftarrow \mathcal{G}$;
2: **for** each $a\_ent$ in $a\_ent\_list$ **do**
3:     $path \leftarrow BFS\_find\_shortest\_path(\mathcal{G}, q\_ent, a\_ent)$;
4:     $L.extend(path)$;
5: **end for**
6: $selected\_triples \leftarrow random\_select(L)$;
7: **for** each $t$ in $selected\_triples$ **do**
8:     $\mathcal{G}'.remove(t)$;
9: **end for**
10: **return** $\mathcal{G}'$;

---

## A.2 Implementation Details

We fine-tune the proposed approach with LoRA. The initial learning rate is $2e-5$, and the sequence is 4096 for all the backbone models. The training epoch is 3, and the batch size is 4. We adopt the AdamW optimizer [21] with a cosine learning scheduler. During the inference, we adopt vLLM [19] to accelerate the reasoning process. All the training and inference experiments are conducted on 2 NVIDIA A800 80G GPUs within 3 hours. Detailed hyperparameters used in our experiments are displayed in Table 6.

## A.3 Prompt for Baselines

In this section, we demonstrate the prompt template and special cases that the SymAgent encounters during its operation. The

## A.4 Case Study

