# OpenReview forum: "SymAgent: A Neural-Symbolic Self-Learning Agent Framework for Complex Reasoning over Knowledge Graphs"
_ACM.org/TheWebConf/2025/Conference — WWW 2025 Oral_

### Official Review · Reviewer_fXa3 · 2024-11-28

**Novelty:** 6
**Technical Quality:** 6

**Review:**

This paper proposes a novel neural-symbolic self-learning agent framework, SymAgent, which autonomously and effectively integrates LLMs and KGs for reasoning tasks on complex knowledge graphs, demonstrating high research quality. The experimental section evaluates the framework on multiple datasets and compares it against several strong baselines, showing that SymAgent outperforms others on various metrics.
Pros:
 1) The combination of large language models (LLMs) and knowledge graphs (KGs) through a self-learning framework for self-improvement is a notable innovation. The introduction of the Agent-Planner and Agent-Executor modules enables the model to dynamically extract symbolic rules from knowledge graphs and perform complex reasoning tasks.
2) The proposed SymAgent framework is original, integrating a self-learning mechanism into neural-symbolic reasoning, which is relatively rare in previous studies. The design of the self-learning framework and the action toolset reflects the authors' original thinking. However, the method in the paper may still be influenced by existing technical approaches.
3) his work is important for improving the accuracy and efficiency of knowledge graph reasoning, especially when dealing with incomplete knowledge graphs.
Cons:
There are also some shortcomings in the paper. For instance, the small image in the upper-left corner of Figure 2 obscures text, and the overall font size is too small to read clearly. Additionally, the discussion on the limitations of the model is insufficient. The paper could explore the challenges the model may face in specific scenarios and suggest future research directions. Furthermore, the scalability of SymAgent in handling larger knowledge graphs is not explicitly addressed in the paper.

**Questions:**

1) The paper mentions the "neural-symbolic agent framework." Could you further elaborate on the specific meaning of this concept?
2) The paper mentions using symbolic rules to guide the reasoning process. Can this improve the interpretability of the model, allowing users to understand the reasoning process?
3) Has the paper considered the limitations of the model in specific scenarios? Could you further discuss these limitations and their impact on the model's performance, as well as suggest potential future research directions?
4) How scalable is SymAgent for large-scale knowledge graphs?

**Reviewer Confidence:**

4: The reviewer is certain that the evaluation is correct and very familiar with the relevant literature

**Scope:**

4: The work is relevant to the Web and to the track, and is of broad interest to the community

---

### Official Review · Reviewer_sYgg · 2024-12-01

**Novelty:** 4
**Technical Quality:** 4

**Review:**

This paper introduces SymAgent, an automatic reasoning framework that combines large language models (LLMs) and knowledge graphs (KGs) to address KG incompleteness through symbolic rules, automatic tool invocation, and self-learning methods.  Experimental results show that SymAgent outperforms or matches existing baseline methods on datasets such as WebQSP, CWQ, and MetaQA-3hop, showcasing its strong capabilities in reasoning tasks. This paper is well-written. However, some of the figures are somewhat cluttered, and the key points are not always highlighted effectively.

Pros：

1.This paper combines knowledge graphs and large language models for collaborative enhancement to address the issue of KG incompleteness, offering a fresh perspective.

2.The paper demonstrates the superior performance of the proposed method through extensive experiments.

3.The paper is well-written.

Cons：
1.The figures are not clear enough; some diagrams, like Figure 2, appear cluttered overall, which affects the reader's understanding of the overall process.

2.The paper does not mention the time and resource costs of the proposed method. This type of model often requires more time for iteration and optimization, limiting its application scenarios.

3.The self-learning framework has significant limitations, especially due to its reliance on LLMs, as their hallucination problem still needs to be addressed. Although the reasoning chains generated by LLMs have been controlled, many unknown factors remain, which is the main cause of performance fluctuations. Therefore, this self-learning module still needs improvement.

4. The appendix seems incomplete, and the paper itself appears unfinished.

**Questions:**

1.Although the appendix of this paper mentions the time and resource consumption of the proposed method, please provide a more detailed comparison of the computational cost, training time, inference time, and resource requirements between this method and the baseline methods. If the additional cost is too high, the authors should consider potential strategies to mitigate these expenses.

2.The paper has conducted an error analysis, which I find very interesting. The paper should provide a more detailed explanation of why the model exhibits different error distributions across different datasets, and what specific characteristics of each dataset contribute to these variations in error distribution?

3.Could the authors provide the missing details in the appendix?

**Reviewer Confidence:**

3: The reviewer is confident but not certain that the evaluation is correct

**Scope:**

3: The work is somewhat relevant to the Web and to the track, and is of narrow interest to a sub-community

---

### Official Review · Reviewer_kziN · 2024-12-02

**Novelty:** 6
**Technical Quality:** 5

**Review:**

This paper presents a novel Neural-Symbolic Self-Learning Agent Framework called SymAgent, which is designed for complex reasoning over KG. SymAgent combines the strengths of knowledge graphs and LLMs, addressing the hallucination issues that often arise with LLMs in complex reasoning tasks. By treating the KG as a dynamic environment, SymAgent transforms complex reasoning tasks into a multi-step interaction process, allowing the KG to play a more integral role in the reasoning process.

## Strengths
- The paper is well-structured, with clear language and logical coherence.
- The proposed SymAgent framework effectively integrates LLMs and KGs for complex reasoning and introduces a self-learning framework that demonstrates promising innovation.
- Experimental results validate the framework's performance.
- Moreover, SymAgent addresses the incompleteness of KGs and the hallucination issue of LLMs, showing potential for application in complex reasoning tasks.

## Weaknesses
1. In Section 5.5, the paper mentions four types of errors and provides statistical analysis but does not offer a detailed examination of each error type.
2. The datasets used in the paper mainly consist of general knowledge graphs and movie KGs. The model's generalization ability across more specialized domains, such as legal knowledge graphs, has not been tested.
3. The paper mentions that during the online exploration phase, the agent optimizes its trajectory using a reward mechanism based on the final result. However, it is unclear whether this final-answer-based reward mechanism adequately captures the quality of the intermediate steps or the reasoning process itself. The current reward mechanism focuses solely on the final answer and overlooks the quality of the intermediate reasoning steps.

**Questions:**

1. Can you compare the experimental results with the ChatGPT O1-preview?
2. The paper mentions that SymAgent addresses the incompleteness of knowledge graphs by retrieving external documents and identifying missing triples. It also tests performance with a retrieval-augmented generation model (RoG). However, could there be semantic inconsistencies that affect the integration of the knowledge graph and, in turn, the accuracy of subsequent reasoning?
3. How does SymAgent compare with retrieval-augmented methods in terms of advantages?

**Reviewer Confidence:**

2: The reviewer is willing to defend the evaluation, but it is likely that the reviewer did not understand parts of the paper

**Scope:**

3: The work is somewhat relevant to the Web and to the track, and is of narrow interest to a sub-community

---

### Official Review · Reviewer_mSzE · 2024-12-03

**Novelty:** 5
**Technical Quality:** 5

**Review:**

In this paper, the authors introduce a novel neural-symbolic agent framework, SymAgent, which achieves collaborative enhancement between Knowledge Graphs (KGs) and Large Language Models (LLMs). The primary motivation stems from limitations in current research: overlooking KG incompleteness and neglecting the implicit logical reasoning structures inherent in KGs. The paper conceptualizes KGs as dynamic environments and transforms complex reasoning tasks into a multi-step interactive process, allowing KGs to participate deeply in reasoning. Additionally, it proposes a self-learning framework that includes online exploration and offline iterative policy updates, enabling the agent to automatically synthesize reasoning trajectories and improve performance. With a relatively weak backbone (i.e., 7B series models), SymAgent demonstrates comparable or better performance than various strong baselines.


Strong Points:
1. The authors conceptualize KGs as dynamic environments, transforming complex reasoning tasks into multi-step interactive processes. This enables KGs to deeply engage in the reasoning workflow, marking a significant departure from treating KGs as static repositories.
2. Experimental evaluations show strong performance, even with weaker LLM backbones (e.g., 7B models), compared to advanced baselines, demonstrating the practical effectiveness of SymAgent.

Weak Points:
1. How the Agent-Planner accurately extracts symbolic rules from KGs and how these rules guide question decomposition remain unclear.
2. It is suggested that the authors should provide an analysis of the limitations of these baselines, which can be used to support the proposed method. This lack of detailed discussion might lead to biased interpretations of SymAgent's performance.

**Questions:**

N.A.

**Reviewer Confidence:**

3: The reviewer is confident but not certain that the evaluation is correct

**Scope:**

3: The work is somewhat relevant to the Web and to the track, and is of narrow interest to a sub-community